# Static posturography as a novel measure of the effects of aging on postural control in dogs

Alejandra Mondino[1], Grant Wagner[1], Katharine Russell[1], Edgar Lobaton[2], Emily Griffith[3], Margaret Gruen[1], B. Duncan X. Lascelles[1,4,5,6], Natasha Jane Olby[1]*

1 Department of Clinical Sciences, College of Veterinary Medicine, North Carolina State University, Raleigh, NC, United States of America, 2 Department of Electrical and Computer Engineering, North Carolina State University, Raleigh, NC, United States of America, 3 Department of Statistics, North Carolina State University, Raleigh, NC, United States of America, 4 Comparative Pain Research and Education Center, College of Veterinary Medicine, North Carolina State University, Raleigh, NC, United States of America, 5 Thurston Arthritis Center, UNC School of Medicine, Chapel Hill, NC, United States of America, 6 Center for Translational Pain Research, Department of Anesthesiology, Duke University, Durham, NC, United States of America

* njolby@ncsu.edu

**Data Availability Statement:** All relevant data are within the paper and its Supporting Information files.

## Abstract

Aging is associated with impairment in postural control in humans. While dogs are a powerful model for the study of aging, the associations between age and postural control in this species have not yet been elucidated. The aims of this work were to establish a reliable protocol to measure center of pressure excursions in standing dogs and to determine age-related changes in postural sway. Data were obtained from 40 healthy adult dogs (Group A) and 28 senior dogs (Group B) during seven trials (within one session of data collection) of quiet standing on a pressure sensitive walkway system. Velocity, acceleration, root mean square, 95% ellipse area, range and frequency revolve were recorded as measures of postural sway. In Group A, reliability was assessed with intraclass correlation, and the effect of morphometric variables was evaluated using linear regression. By means of stepwise linear regression we determined that root mean square overall and acceleration in the craniocaudal direction were the best variables able to discriminate between Group A and Group B. The relationship between these two center-of-pressure (COP) measures and the dogs' fractional lifespan was examined in both groups and the role of pain and proprioceptive deficits was evaluated in Group B. All measures except for frequency revolve showed good to excellent reliability. Weight, height and length were correlated with most of the measures. Fractional lifespan impacted postural control in Group B but not Group A. Joint pain and its interaction with proprioceptive deficits influence postural sway especially in the acceleration in the craniocaudal direction, while fractional lifespan was most important in the overall COP displacement. In conclusion, our study found that pressure sensitive walkway systems are a reliable tool to evaluate postural sway in dogs; and that postural sway is affected by morphometric parameters and increases with age and joint pain.

**Funding:** The authors received no specific funding for this work.

**Competing interests:** The authors have declared that no competing interests exist.

## Introduction

Aging is associated with impairment in postural control in humans due to a decline in the motor [1] and sensory systems [2, 3]. Impairments in the former involve a reduction in muscle mass and strength [1] and the co-activation of antagonist muscles [4]. Changes in the sensory systems involve reduced proprioception, vestibular and visual functions [3, 5, 6]. It has been demonstrated that deficiencies in postural control and balance are associated with frailty, which, in turn, predicts adverse outcomes such as disability and mortality [7–10]. Similarly, older dogs experience a reduction in muscle mass, and a decline in sensory functions such as hearing and vision loss [11–13]. Additionally, dogs have a higher prevalence of idiopathic vestibular disease than younger dogs [14]. However, to the authors' knowledge, the effect of aging on postural control in dogs has not yet been elucidated. Dogs are a powerful model for the study of aging for several reasons [15]. First of all, they are considered family members and share the living environment with their owners, which exposes them to the same stimuli and potential toxins [16]. Secondly, chronic, and age-related diseases are very well characterized in dogs, and prevalent in an aged population of dogs who are supported by an advanced health system that allows the diagnosis and treatment of disease in a manner analogous to people [16, 17]. In addition, many of the same chronic conditions occur in both dogs and humans [16]. Establishing methods for evaluating postural control in dogs would contribute to the utility of dogs as a model of aging. A useful way to evaluate postural control is static posturography, an evaluation of the postural sway during quiet standing. This is usually assessed using various measurements of the motion of the center-of-pressure (COP). The COP is the two-dimensional projection of the center of gravity and reflects the orientation and movements of the body. It changes constantly, creating a COP or sway path [18–20]. The COP displacement can be determined by means of pressure sensitive walkway systems (PSW) or force-plates. While force-plates are the gold-standard device for these measurements, PSW are lightweight, and easily transportable, making them more suitable for clinical settings [18–21].

In the past few years, the number of posturographic studies in dogs has increased, but most studies have focused on the effect of orthopedic or spinal cord diseases on COP displacement during walking; none have focused on static posturography [19, 22–24]. Additionally, no study has yet quantified the reliability of the COP measurements obtained using a PSW in standing dogs or evaluated the effect of morphological variables such as height or weight on the COP measurements. In humans and horses, it has been demonstrated that those variables can influence postural sway [25–27]. In dogs, determining if morphological features influence the outcome of posturography measurements is critical due to the large variation in body shapes and sizes in this species. Therefore, the aims of this work were to establish a protocol to measure COP excursions in standing dogs; to evaluate the influence of morphological characteristics on these measurements; and to determine age-related changes in postural sway in the dogs using a wide range of COP excursion measurements.

## Materials and methods

### Dogs

All protocols were reviewed and approved by the North Carolina State University Institutional Animal Care and Use Committee and owners reviewed and signed and informed consent form.

In order to optimize a COP measurement protocol and to evaluate how morphological features can affect postural sway, client-owned dogs were recruited through the North Carolina State University College of Veterinary Medicine. Dogs were selected from a range of different

breeds, ages and sizes. All dogs were required to weigh more than 5 kg for their COP to be detected by the PSW. Due to the effect of body size on lifespan in dogs [28, 29], each dog's lifespan was estimated using the formula described by Greer *et al.* (2007) [30] which takes into consideration the height and weight of the dog. The fractional lifespan was calculated as the dog's actual age divided by their estimated lifespan; dogs older than 1 year and fractional lifespan $\leq 0.75$ were included in Group A (adult) and dogs with fractional lifespan $> 0.75$ were included in Group B (seniors) [31]. To be included, dogs in Group A had to have a normal orthopedic and neurological examination, while dogs in Group B needed to be able to walk independently.

## Clinical evaluation

Clinical evaluation consisted of physical, neurologic, and orthopedic examinations that were recorded using standard clinical evaluation sheets (S1 File). During the physical examination, dog's height to the dorsal border of the scapula (withers) and weight were measured [32]. The neurological examination included subjective gait and posture evaluations, assessment of conscious proprioception (CP) by means of placing responses, spinal reflexes and cranial nerve examination as well as evaluation for head or spinal pain [33]. Regarding CP, dogs without any CP deficits were classified as "Normal", while dogs with $\geq 1$ limb with decreased or absent response were categorized as "Abnormal". The orthopedic examination included gait evaluation, evaluation of muscle mass and symmetry, and passive flexion/extension of each joint to assess range of motion and joint pain, and palpation of each joint to assess thickening and/or effusion. All appendicular joints were assessed and scored, with the manus and pes joints each being considered a single site. Joint pain was quantified using an established scale [34] by grading each joint during manipulation from 0 to 4 as follows: 0: Does not notice; 1: Orients to site, Does not resist or mild resistance; 2: Orients to site, slight objection to manipulation; 3: Withdraws from manipulation, may vocalize, may turn to guard area and 4: Tries to escape/prevent manipulation, may bite or show aggression. The score from each joint was summed to calculate the total joint pain on a scale of 0–64. We also evaluated the owners' perception of their dogs' pain and function/mobility by means of two different questionnaires: Canine Brief Pain Inventory (CBPI) [35] and Liverpool Osteoarthritis in Dogs (LOAD) [36]. These questionnaires are not direct measures of pain, rather of how joint pain impacts function and mobility. However, the CBPI can be divided into two subscales, a Pain Severity subscale (intended to measure the severity of a dog's pain evident to owners) and a Pain interference subscale (intended to measure how pain interferes in dog's daily activities). The first one is obtained by the average of the responses of the first 4 questions and the second one by the average of the last 6 questions [37]. Higher scores in these questionnaires indicate higher severity or interference of pain. The CBPI Pain Severity and Interference subscale range is 0–10 LOAD range is 0–52.

## Equipment

The PSW used in this study was the 7100 QL Virtual Sensor 4 Mat System, (Tekscan, Boston, MA) with I-Scan software (version 5.231; Tekscan, Boston, MA). Details of this system can be found elsewhere [38]. Prior to data acquisition, the sensors were equilibrated and calibrated as per the manufacturer's specifications. The frequency of sampling was 67 frames per second. The duration of the trials was 8 seconds (536 frames). Between trials, dogs were allowed to move before re-establishing the standing position so that subsequent trials were not simply a continuation of the previous stance position. A handler was always placed on the right side of the dog, holding a leash, which was hanging loosely from the dog. Each trial was video

recorded. At least 10 trials were collected from any given dog. Following acquisition, videos were reviewed and only the frames where the dogs were standing square, completely still and looking forward were considered acceptable for analysis. To have a consistent number and duration of trials in all dogs, from all the frames considered acceptable by visual inspection of the videos, we selected 7 periods of 70 acceptable frames by means of a Matlab script (Matlab, version 2021a; The MathWorks INC, Natick, MA). Dogs with less than 7 periods of 70 acceptable frames were excluded from the study. The number of periods and frames were chosen to maximize the amount of data analyzed without having to exclude more than 20% of the tested dogs. If at least 7 trials had periods of 70 acceptable frames, the script selected one period of each trial in the order they were recorded, if not, it sampled the trials available more than once and selected additional non-overlapping periods.

## COP variables calculation

We used a Matlab script (Matlab, version 2021a; The MathWorks INC, Natick, MA) to calculate 16 different COP time-domain measures relating to postural sway. A low pass 5 Hz filter was applied to the data obtained from the PSW before the analysis [20]. Each variable was calculated for each period of 70 frames and then averaged to obtain one value per each dog. The detailed calculation of each variable can be found in Prieto *et al*. 1996 [20] and Bickley *et al*. 2019 [18]. These measures can be divided into three main categories: distance, area, and hybrid parameters [20]. We calculated the following distance parameters: craniocaudal (CC), mediolateral (ML), and overall (Ov) velocity (Vel); CC, ML, and Ov acceleration (Acc), CC and ML range and CC, ML and Ov root mean square (RMS) distance of the mean COP. The Vel in the CC and ML directions refers to the total distance traveled by the COP in one second in the CC and ML directions, while the Ov Vel takes into consideration the total COP displacement over time regardless of its direction. Acc averages the changes in Vel from the COP position at each frame to the next using the absolute value, so decelerations do not cancel accelerations. Range represents the maximal distance travelled by the COP each direction. The RMS quantifies the variability of the COP position, and it was calculated as follows: the mean COP was defined by the arithmetic means of the CC and ML positions during the trial; then the resultant distance (RD) time series (i.e., the vector distance from the mean COP to each pair of COP positions in the CC and ML time series) were calculated. The RMS Ov was computed as the RMS of the RD time series. The RMS CC and ML were calculated as the standard deviation of the CC and ML time series respectively. We also calculated an area parameter, the 95% confidence ellipse area. Finally, we calculated the following hybrid measures which are dependent on both, the distance and area traveled by the COP: sway area and CC, ML, and Ov frequency revolve (Freq). Sway area was determined by the average of the area of a triangle generated by two consecutive CC and ML data points (COP position) and the mean COP per second. Freq Ov refers to the revolutions per second that the COP travelled around a circle with the radius of the mean distance of the COP; while Freq CC or ML are the frequency in Hz of a sinusoidal oscillation with an average value of the mean distance and a total length of the distance traveled by the COP in the CC or ML directions respectively [20]. We can also group these variables in the ones that give information about the length of the sway path (Vel or Acc) and the ones that depend on the extension of the COP displacement (range, RMS, 95% ellipse and sway area). Fig 1 shows a schematic of some of the variables.

## Statistical analysis

Statistical analyses were performed using JMP Pro, Version 15.2.0. SAS Institute Inc., Cary, NC and IBM SPSS Statistics, Version 27.0. IBM Corp. Armonk, NY. Normality was evaluated

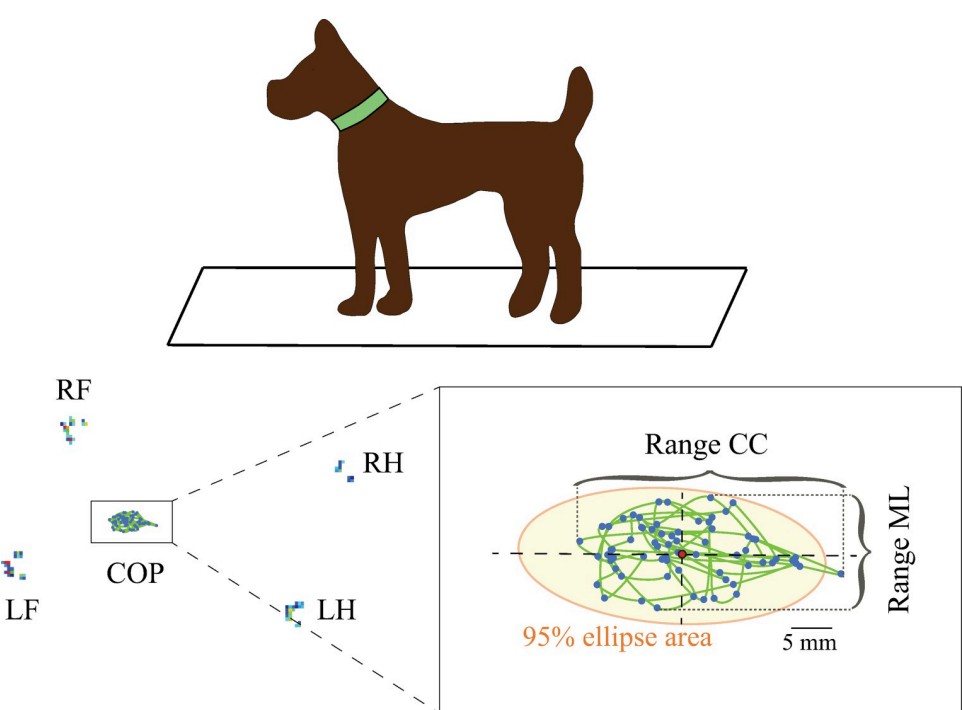

**Fig 1. COP displacement evaluation in dogs.** Top: Schematic representation of a dog standing still on a pressure sensitive walk-way system. Left Bottom: Location of each limb and COP trajectory during one trial. Right Bottom: Magnification of the COP trajectory. Blue dots represent the location of the COP at each frame, and the red dot represents the centroid or Mean COP. The brackets correspond to the range in the CC and ML directions and the orange ellipse represents the 95% ellipse area of the COP.

by a Shapiro-Wilk test. None of the COP variables were normally distributed, therefore, data were log transformed to achieve normality. Additionally, total joint pain and CBPI subscales scores were also not normally distributed and because some dogs had values of zero, transformation to achieve normality was performed by log(x+1). Analyses were performed using transformed data. Reliability of the COP measurements was calculated by intraclass correlation (ICC) analysis using a two-way random effects, absolute agreement, multiple measurements i.e., ICC (2,$k$), $k$ = 7. The selection of the model and type of ICC was based on the recommendations of Koo and Li (2016) [39]. We selected the ICC type "mean of $k$ measures" because our intended protocol averages the results of several trials, and not using just one trial. This decision was based in several studies in humans that have demonstrated that an average of a minimum of two trials is needed to obtain a reliable measure of postural stability [40–42]. Values greater than 0.9 were considered indicators of excellent reliability, between 0.75 and 0.9 of good reliability, between 0.5 and 0.75 of moderate reliability, and less than 0.5 of poor reliability [39]. Single linear regressions were used to calculate the correlation between the morphometric variables and the COP measurements in the group of younger dogs. We wanted to evaluate several COP measurements in dogs as a preliminary exploration to determine which would be of most value for future studies. However, those measurements have shown to be correlated in humans and can, therefore, be redundant. A forward stepwise regression was used to examine which of the COP variables had the greatest power to discriminate between younger and older dogs. At each step, variables were added based on p-values, and the BIC was used to set a limit on the total number of variables included in the final model. Once the model was selected, we calculated the area under the curve (AUC) of the Receiver Operating

Characteristic (ROC) curve to evaluate its performance. Additionally, we evaluated the effect of fractional life span on the selected variables in each group and we performed a multivariate analysis including "group", "fractional lifespan" and their interaction to evaluate differences in the slope of each group. Furthermore, we calculated the correlation between the selected COP measurements and CP, total joint pain, CBPI subscales and LOAD score. We performed a stepwise regression in each of the selected COP measurements including all variables that showed a significant effect to determine the best model able to predict COP displacement. We evaluated this in dogs by means of a multivariate correlation. For all analyses, probability (p) value <0.05 was considered statistically significant. Because this is an exploratory study, using a wide range of COP measurements, we did not correct p values for multiple comparisons. The raw data used in this article is provided in S2 File.

## Results

### Group A

**Demographics.** This group consisted of 49 dogs younger than 75% of their calculated life span. Eight dogs were excluded because they didn't have enough acceptable trials. Also, an additional dog was excluded from the regression analyses for being an outlier. The remaining 40 dogs consisted of 19 females (47.5%; 3 were intact and 16 spayed), and 21 males (52.5%; 5 intact and 16 castrated). The mean age was 4.52 ± 2.20 years (range: 1.04 to 9.08). The mean weight was 21.04 ± 11.07 kg, and the mean height was 51.47 ± 14.00 cm. The median and inter-quartile range of body condition score (BCS) was 5, (range 4–9), 62.5% of the dogs had an ideal BCS (score 4–5). Every dog in this group had a normal neurological and orthopedic examination. The dogs belonged to 13 different breeds with 2 dogs (5%) being mix breed. The most frequent breed was border collie (n = 7, 17.5%), followed by Siberian husky (n = 5, 12.5%). Other breeds were German shepherds and American pitbull (n = 4, 10% each), golden retriever, Labrador retriever and Jack Russell terrier (n = 3, 7.5% each), Scottish terrier, and Boston terrier (n = 2, 5% each) and 1 (2.5%) American bulldog, beagle, chihuahua, Maltese terrier, and Scotch collie. This group was used to calculate the reliability of the COP measurements and the influence of morphometric variables on them.

**Reliability of COP measurements.** The results of intra-test reliability of the COP measurements are reported in Table 1. VelCC, VelOv, AccCC, VelCC, and sway area showed an excellent reliability when using the average of 7 different trials. Vel ML and 95% ellipse had a good to excellent reliability and Range CC, Range ML, RMSCC, RMS ML, RMS Ov, and Acc ML had moderate to good reliability. Frequency measurements in all directions showed a poor to good reliability and were therefore excluded from further analysis.

**Influence of morphometric variables on the COP measurements.** Most COP measurements showed a positive correlation with weight, height and length while Acc ML had a negative correlation with weight and length (Table 2). From the three measurements length was the one that showed higher correlation coefficients with COP measures. In this group of dogs, the three variables (weight, height and length) were highly positively correlated with each other (r = 0.897, p = <0.001 for weight and height, r = 0.812, p = <0.001 for weight and length and r = 0.857, p = <0.001 for height and length). Because of this, and as length had the highest correlation, we included only length as a covariate in subsequent analyses.

### Group B

**Demographics.** This group consisted of 27 dogs older than 75% of their calculated life-span. Three dogs were excluded because they didn't have enough acceptable trials. The remaining 24 dogs consisted of 6 males (25%; all castrated) and 18 females (75%; all spayed).

**Table 1. Reliability of COP measurements.**

| | | 95% Confidence Interval | |
| --- | --- | --- | --- |
| | ICC | Lower bound | Upper bound |
| Vel CC | 0.951 | 0.924 | 0.971 |
| Vel ML | 0.899 | 0.844 | 0.94 |
| Vel Ov | 0.953 | 0.927 | 0.972 |
| Acc CC | 0.952 | 0.926 | 0.972 |
| Acc ML | 0.833 | 0.741 | 0.901 |
| Acc Ov | 0.943 | 0.912 | 0.966 |
| Range CC | 0.82 | 0.721 | 0.893 |
| Range ML | 0.806 | 0.699 | 0.885 |
| RMS CC | 0.773 | 0.648 | 0.865 |
| RMS ML | 0.789 | 0.673 | 0.874 |
| RMS Ov | 0.798 | 0.686 | 0.88 |
| Sway area | 0.946 | 0.916 | 0.968 |
| 95% ellipse | 0.905 | 0.854 | 0.944 |
| Freq CC | 0.522 | 0.261 | 0.761 |
| Freq ML | 0.611 | 0.397 | 0.769 |
| Freq Ov | 0.622 | 0.415 | 0.775 |

Results of Intraclass correlation analysis using SPSS multiple-rating, absolute agreement, 2-way random effects model; ICC (2,$k$). This analysis was performed with the data of adult dogs (Group A).

The mean age was 12.20 ± 1.53 years (range 9.8–15.60). The mean weight was 21.29 ± 11.27 kg, the mean height 51.07 ± 14.44 cm and the mean length 56.48 ± 12.54 cm. The median BCS was 5 (range: 2–7), 54% of the dogs had an ideal BCS (score 4–5). Dogs belonged to 12 different breeds and 4 (16.7%) were mix breed dogs. The most frequently represented breed was American pitbull (n = 4, 16.7%) followed by Labrador retriever (n = 3, 12.5%). Other breeds included Australian cattle dog, beagle, and Jack Russell terrier (n = 2, 8.3% each); and border

**Table 2. Correlation between morphometric variables and postural sway in younger dogs.**

| | Weight | | Height | | Length | |
| --- | --- | --- | --- | --- | --- | --- |
| | r | p value | r | p value | r | p value |
| Vel CC | 0.370 | **0.019**\* | 0.402 | **0.011**\* | 0.447 | **0.005**\* |
| Vel ML | -0.071 | 0.661 | -0.007 | 0.967 | 0.080 | 0.609 |
| Vel Ov | 0.320 | **0.044**\* | 0.364 | **0.023**\* | 0.428 | **0.008**\* |
| Acc CC | 0.372 | **0.018**\* | 0.398 | **0.012**\* | 0.407 | **0.012**\* |
| Acc ML | -0.333 | **0.035**\* | -0.243 | 0.135 | -0.251 | **0.134**\* |
| Acc Ov | 0.218 | 0.177 | 0.270 | 0.097 | 0.288 | 0.084 |
| Range CC | 0.378 | **0.016**\* | 0.367 | **0.021**\* | 0.460 | **0.004**\* |
| Range ML | 0.312 | **0.050**\* | 0.308 | 0.056 | 0.567 | **<0.001**\* |
| RMS CC | 0.403 | **0.010**\* | 0.387 | **0.015**\* | 0.500 | **0.002**\* |
| RMS ML | 0.375 | **0.017**\* | 0.371 | **0.020**\* | 0.603 | **<0.001**\* |
| RMS Ov | 0.396 | **0.011**\* | 0.383 | **0.016**\* | 0.552 | **<0.001**\* |
| Sway Area | 0.395 | **0.012**\* | 0.430 | **0.006**\* | 0.568 | **<0.001**\* |
| 95% ellipse | 0.440 | **0.004**\* | 0.462 | **0.003**\* | 0.619 | **<0.001** |

Linear regression analysis results showing the correlation between morphometric variables and postural sway in adult dogs (Group A). n = 40.

collie, boxer, dachshund, German shorthaired pointer, golden retriever, Great Dane, and Irish setter (n = 1, 4.2% each). Of note, weight, height and length were not significantly different between Group A and Group B, (t = 1.99, p = 0.914, t = 1.99, p = 0.931 and t = 2.00, p = 0.99, respectively).

**Pain and owner-assessed function evaluation.** Twenty-three owners completed the CBPI and nineteen completed the LOAD questionnaire. The median Pain Severity subscale of CBPI score was 1 (range 0–5.5; higher scores indicate more pain) and the median total joint pain was 4 (range 0–17). Four dogs (14.28%) did not show any joint pain. The median Pain Interference subscale of the CBPI was 0.7 (range 0–6.8; higher scores indicate more pain interference). Median LOAD score was 14 (range 4–29; higher scores indicated worse function). We also studied the correlation between pain and owner-assessed function evaluation and fraction of lifespan (Fig 2). Of these assessments (CBPI, total joint pain and LOAD), all except for total joint pain showed a positive correlation with fractional lifespan.

**Proprioception.** None of the dogs had absent proprioceptive placing in any limb, but 9 dogs (37.5%) had reduced response in at least 1 limb. Within Group B, dogs with abnormal proprioception had a higher fractional lifespan than normal dogs (1.08 ± 0.06 and 0.94 ± 0.06 respectively, t = 3.58, p = 0.002).

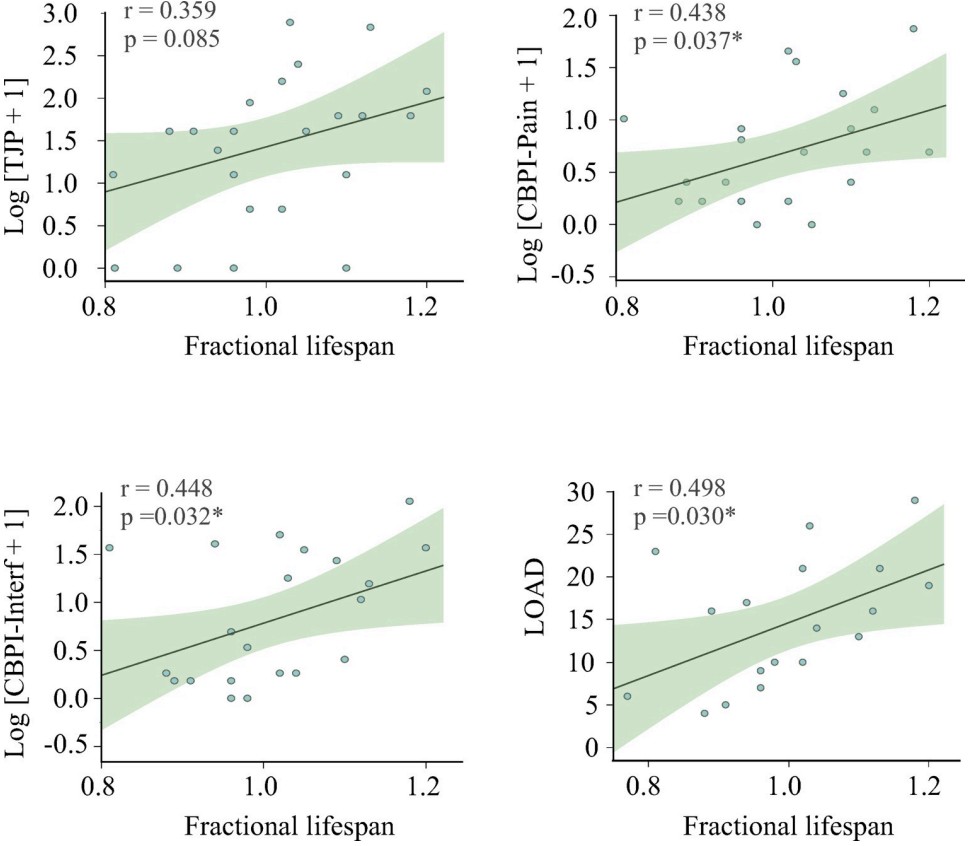

**Fig 2. Correlation between fractional lifespan, pain and owner-assessed function evaluation.** Pain was assessed by orthopedic examination (total joint pain) and by the Pain Severity subscale of the owner-based questionnaire, the Canine Brief Pain Inventory (CBPI). Additionally, owners' perception of dogs' function was evaluated by means of the Pain Interference subscale of the CBPI and the Liverpool Osteoarthritis in Dogs (LOAD) questionnaire. During the orthopedic examination the pain of each joint was graded according to the scale described by Knazovicky *et al.* (2016) [34] and total joint pain was calculated as the sum of each joint pain score.

**Table 3. Multivariate analysis showing correlations between COP measurements in all dogs.**

|  | Vel CC | Vel ML | Vel Ov | Acc CC | Acc ML | Acc Ov | Range CC | Range ML | RMS CC | RMS ML | RMS Ov | Sway Area | 95% ellipse |
|---|---|---|---|---|---|---|---|---|---|---|---|---|---|
| Vel CC |  | 0.638 | 0.986 | 0.974 | 0.496 | 0.942 | 0.840 | 0.555 | 0.824 | 0.519 | 0.706 | 0.813 | 0.751 |
| Vel ML | 0.638 |  | 0.747 | 0.597 | 0.883 | 0.755 | 0.651 | 0.746 | 0.669 | 0.667 | 0.717 | 0.761 | 0.724 |
| Vel Ov | 0.986 | 0.747 |  | 0.955 | 0.589 | 0.958 | 0.843 | 0.634 | 0.837 | 0.594 | 0.780 | 0.859 | 0.797 |
| Acc CC | 0.974 | 0.597 | 0.955 |  | 0.501 | 0.958 | 0.726 | 0.450 | 0.707 | 0.420 | 0.641 | 0.728 | 0.642 |
| Acc ML | 0.492 | 0.883 | 0.589 | 0.501 |  | 0.709 | 0.443 | 0.397 | 0.417 | 0.305 | 0.406 | 0.452 | 0.386 |
| Acc Ov | 0.942 | 0.755 | 0.958 | 0.958 | 0.709 |  | 0.749 | 0.504 | 0.720 | 0.448 | 0.664 | 0.741 | 0.655 |
| Range CC | 0.840 | 0.651 | 0.843 | 0.726 | 0.443 | 0.749 |  | 0.721 | 0.982 | 0.670 | 0.929 | 0.859 | 0.871 |
| Range ML | 0.555 | 0.746 | 0.634 | 0.450 | 0.397 | 0.504 | 0.721 |  | 0.770 | 0.980 | 0.897 | 0.887 | 0.915 |
| RMS CC | 0.824 | 0.669 | 0.837 | 0.707 | 0.417 | 0.720 | 0.982 | 0.767 |  | 0.736 | 0.965 | 0.896 | 0.913 |
| RMS ML | 0.519 | 0.677 | 0.594 | 0.420 | 0.305 | 0.448 | 0.670 | 0.980 | 0.736 |  | 0.880 | 0.877 | 0.908 |
| RMS Ov | 0.707 | 0.717 | 0.780 | 0.641 | 0.406 | 0.664 | 0.929 | 0.897 | 0.965 | 0.880 |  | 0.940 | 0.964 |
| Sway Area | 0.813 | 0.761 | 0.859 | 0.728 | 0.452 | 0.741 | 0.859 | 0.997 | 0.896 | 0.877 | 0.904 |  | 0.978 |
| 95% ellipse | 0.751 | 0.724 | 0.797 | 0.642 | 0.386 | 0.655 | 0.871 | 0.915 | 0.913 | 0.908 | 0.964 | 0.978 |  |

Correlations were estimated by Row-wise method, and correlation values are shown at the bottom right corner of each scatterplot. For this analysis, we included all the dogs used in the study (Group A + Group B).

**Correlation between COP variables.** A multivariate analysis pooling the data of both group of dogs for all the measurements, showed that all variables were significantly correlated ($p < 0.05$). Highest correlations were found between variables at the same direction, in particular between Vel and Acc; range and RMS; sway area and 95% ellipse area; and 95% ellipse area and RMS. Lowest correlations were found for Acc and Vel with all the other variables (Table 3).

**Older dogs showed a higher COP displacement than younger dogs.** The median and ranges of each COP variable in younger and older dogs are shown in Table 4. All parameters were higher in the older dogs. A stepwise regression analysis including all COP variables as possible predictors of the group (i.e. younger vs. older) revealed that approximately 23% of the variance was explained by two independent variables; RMS Ov and Acc CC ($R^2 = 0.23$, $\chi^2(2, N = 64) = 19.31$, $p = <0.0001$). The model including these two variables had an excellent accuracy to discriminate between both groups, with an area under the ROC curve of 0.80.

**Table 4. Median and range values of each COP variable in younger and older dogs.**

| COP variable | Younger (n = 40) | Older (n = 24) |
|---|---|---|
| Vel CC (mm/s) | 23.18 (7.61–53.70) | 24.81 (13.94–55.92) |
| Vel ML (mm/s) | 9.50 (4.58–17.81) | 11.36 (7.59–24.61) |
| Vel Ov (mm/s) | 26.67 (9.69–58.20) | 30.02 (17.63–63.86) |
| Acc CC (mm/s$^2$) | 526.28 (157.82–1376.19) | 565.70 (247.72–1148.12) |
| Acc ML (mm/s$^2$) | 184.45 (93.73–344.82) | 216.92 (109.56–580.63) |
| Acc Ov (mm/s$^2$) | 108.62 (38.03–232.92) | 126.54 (63.28–236.98) |
| Range CC (mm) | 8.06 (3.13–14.82) | 7.39 (5.84–23.09) |
| Range ML (mm) | 4.17 (2.07–9.91) | 5.65 (2.93–11.33) |
| RMS CC (mm) | 1.98 (0.70–4.05) | 2.47 (1.59–5.59) |
| RMS ML (mm) | 1.14 (0.48–2.55) | 1.60 (0.74–3.41) |
| RMS Ov (mm) | 2.36 (0.88–5.06) | 3.09 (1.83–6.56) |
| 95% ellipse (mm$^2$) | 37.00 (4.58–193.89) | 61.55 (21.07–271.44) |
| Sway area (mm$^2$) | 14.42 (1.96–68.46) | 22.07 (9.60–102.88) |

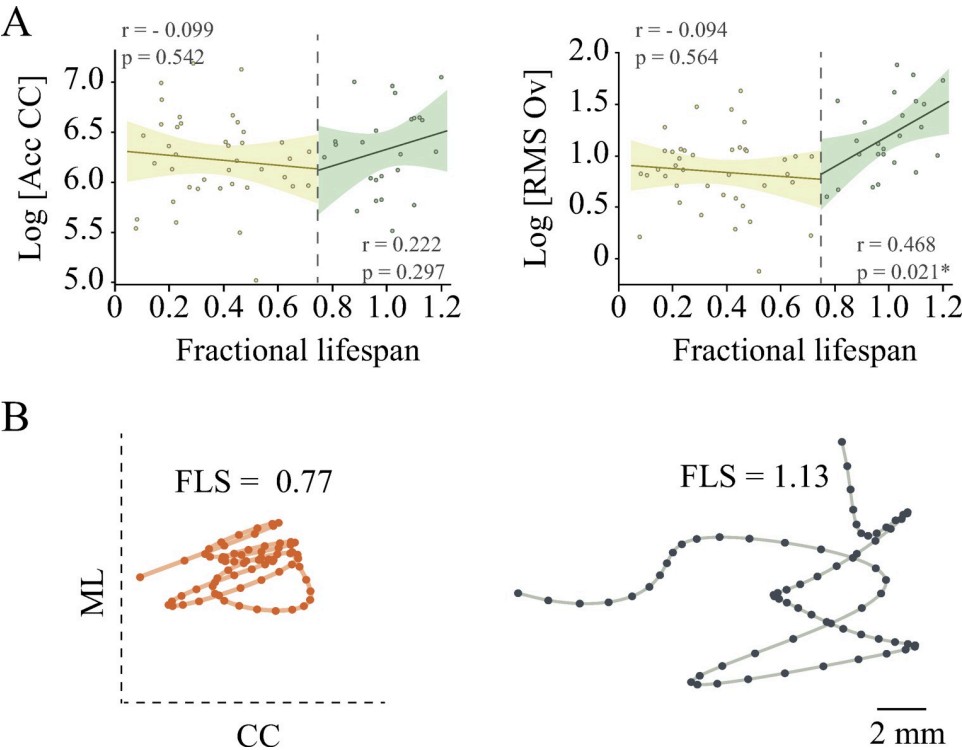

**Fig 3. Correlation between fraction of lifespan and COP measurements in younger and older dogs.** The linear regression between fraction of lifespan and the two COP measurements was evaluated for each group of dogs independently (A). For easier visualization both are shown in the same graph separated by a black dashed line. At the bottom of the figure (B) there are two examples of COP displacement after 5 Hz low pass filtering in two senior dogs with extreme fractional lifespan values.

Since all variables are highly correlated, in order to avoid redundancy, we used only these variables for further analyses. We were also interested in whether postural stability was associated with fractional lifespan in dogs who had not reached their senior stage and were orthopedically and neurologically healthy. We therefore examined the relationship between fractional lifespan and COP displacement in both groups independently. The results are provided in Fig 3. Fractional lifespan was not correlated with Acc CC or RMS Ov in younger dogs (r = - 0.099 p = 0.542 and r = - 0.094, p = 0.564 respectively) or with Acc CC in older dogs (r = 0.222, p = 0.297), but it showed a positive correlation with RMS Ov in Group B (r = 0.468 p = 0.021). Slopes for RMS Ov, but not for Acc CC were different between the two groups (p = 0.021 and 0.233 respectively). We included length as a covariate in this analysis and we found that it did not affect these results (Table 5). Of interest, length was not significantly correlated with any of the COP variables analyzed in the older dogs.

**Table 5. Relationship between COP variables and FLS including length as a covariate in the model.**

| | Group A—Younger | | | | | Group B—Older | | | | |
|---|---|---|---|---|---|---|---|---|---|---|
| | Model | Frac. lifespan | | Length | | Model | Frac. lifespan | | Length | |
| Variables | $R^2$ | Std β | p | Std β | p | $R^2$ | Std β | p | Std β | p |
| Acc CC | 0.187 | -0.145 | 0.354 | 0.402 | 0.014* | 0.062 | 0.257 | 0.257 | -0.121 | 0.587 |
| RMS Ov | 0.316 | -0.103 | 0.472 | 0.549 | <0.001* | 0.231 | 0.500 | 0.021* | -0.11 | 0.580 |

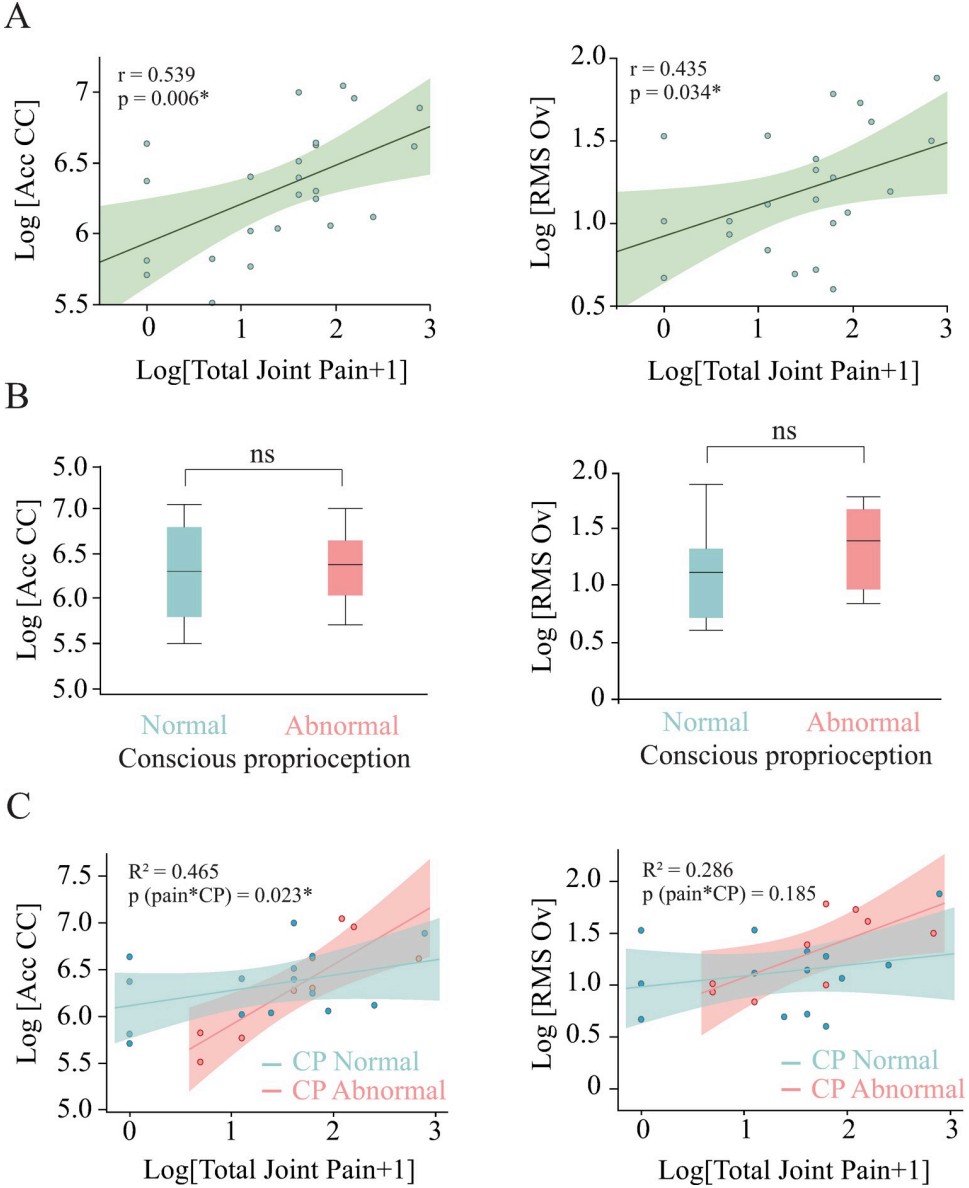

**Fig 4. Relationship between postural stability and, total joint pain and conscious proprioception.** A. Correlation between total joint pain and Acc CC (left) and RMS Ov (right) in older dogs. B. Differences in Acc CC (left) and RMS Ov (right) between older dogs with normal and abnormal proprioception. C. Relationship between total joint pain and Acc CC (left) and RMS Ov (right) for dogs with normal and abnormal proprioception. The effect of the interaction total joint pain * CP was evaluated and p values are shown in the figure.

**Effect of Pain and proprioceptive deficits on postural stability.** Total joint pain was positively correlated with RMS Ov and Acc CC (Fig 4A). We evaluated the owners' perception of their dogs' pain by means of the Pain Severity subscale of the CBPI and found no correlation between the Pain Severity subscale and RMS Ov or Acc CC ($p > 0.05$). Additionally, we measured the owner's perception of mobility of their dogs by means of the Pain Interference subscale of the CBPI and the LOAD questionnaire. No significant correlation was found between these scores and the COP measurements ($p > 0.05$).

We did not see any differences between dogs with normal and abnormal CP for RMS Ov ($t = 2.07$, $p = 0.220$) or Acc CC ($t = 2.07$, $p = 0.992$) (Fig 4B). We expected dogs with a

**Table 6. Stepwise regression model for RMS Ov and Acc CC.**

| RMS Ov. | | | | |
|---|---|---|---|---|
| Predictors | $R^2$ | $R^2$ change | Std β coefficients | p value |
| Fractional life span | 0.219 | | 0.468 | 0.021 |
| Acc. CC | | | | |
| Predictors | $R^2$ | $R^2$ change | Std β coefficients | p value |
| Total joint pain | 0.290 | | 0.793 | <0.001 |
| Total joint pain*CP (Abnormal) | 0.465 | 0.175 | 0.461 | 0.023 |
| CP (Abnormal) | 0.465 | | -0.189 | 0.278 |

combination of CP deficits and joint pain to have worse postural balance than dogs with just one of the two, therefore, we also evaluated the effect of the interaction between total joint pain and CP deficits, and we found that this interaction had a significant effect on Acc CC but not in RMS Ov (Fig 4C). To further evaluate this interaction, we evaluated the correlation between Acc CC with total joint pain for each CP group individually, joint pain was correlated with Acc CC only in the group with abnormal proprioception (r = 0.849, p = 0.004, Vs. r = 0.401, p = 0.138).

A stepwise regression analysis including significant variables as possible predictors of RMS Ov and Acc CC (i.e. fractional life span, total joint pain and the interaction between total joint pain and CP deficits) revealed that RMS Ov was dependent only on fractional life span which explained 22% of the variance while 46.5% of the variance of Acc CC was explained by total joint pain and its interaction with CP deficits Table 6.

## Discussion

In this study we demonstrated that most COP measurements showed good reliability within a session of data collection, except for frequency revolve measurements, and that morphometric variables should be factored into analyses as a covariate. Postural sway increases with fractional lifespan in senior, but not adult dogs. Joint pain, especially in dogs with proprioceptive deficits, increases the acceleration of COP movement in the CC direction while the overall displacement of the COP is influenced mainly by fractional lifespan.

### Reliability of the COP measurements during quiet standing

This study has demonstrated that is it possible to measure the COP displacement in dogs during quiet standing with most of the COP measurements having a moderate to excellent intrasession reliability when averaging 7 trials of 70 frames with a frequency sample of 67 frames/s. Vel CC and Vel Ov as well as Acc CC and Acc Ov showed the highest reliability while frequency measurements showed the lowest reliability. Similar results have been found in humans where velocity showed the highest reliability and other measurements had similar intra-session reliability to the ones found in this study [40, 43, 44]. It is worth noting that increasing the number or the duration of the trials could improve the reliability of the measurements. However, obtaining successful trials in which dogs stay completely still and remain looking forward is not an easy task and obtaining additional trials could be challenging. Additionally, we found that older dogs are prone to fatigue if they stand for long periods of time, limiting the possibility of accomplishing extended trials.

### Influence of morphometric parameters

We found that COP measurements are correlated with weight, height and length; therefore, when comparing groups of dogs, it is necessary to include dogs with similar body size and

shape in each group, or to incorporate at least one of these morphometric variables as a covariate in the statistical analysis. In our dogs, these three variables (weight, height and length) were correlated with COP measurements. With respect to weight, most dogs in our population had an ideal BCS, and thus, our results might not be reproducible in overweight dogs. Due to the low variability of BCS in the dogs in this study, we did not evaluate the effect of BCS in COP measurements. Further studies need to evaluate how morphometric variables impact postural sway in dogs with different BCS. In humans, it has also been demonstrated that anthropometric parameters affect postural stability. Most of these parameters have been shown to have a positive correlation with morphometric variables [26, 45]. However, Chiari *et al.* (2002) found that ML parameters were negatively correlated with the base of support area [26]. Additionally, in horses a negative correlation was also found between Vel CC and ML frequency and body size [25]. In that paper, authors measured height, length, width and mass, and by principal component analysis they consolidated the variables in one single "size" variable. In our study, length seemed to be the most relevant parameter for postural stability, however, all three morphometric variables were highly correlated and considering at least one of them in these types of analyses might be enough for future studies. We found that the effect of morphometric parameters on COP measurements was different in each group of dogs. While all three morphometric variables were correlated with COP displacement in Group A, they were not in Group B. These results suggest that the effect of morphometric variables is different at different life stages which could indicate that aging may have different impact in different sizes of dogs; further studies are needed to evaluate this.

## Postural sway increases with aging

We observed a higher displacement of COP in older dogs in comparison with younger dogs, and we found that a model including Acc CC and RMS Ov is the best to discriminate between the two groups. In order to establish normal ranges, ICC, and the relationship between postural sway measurements and morphometric parameters, the dogs in Group A had to have a normal orthopedic and neurologic examination. However, one of the consequences of aging is that many comorbidities develop and the majority (85.7%) of the dogs in Group B had some degree of joint pain, and 37.5% showed proprioceptive deficits. We therefore evaluated the effect of the fractional lifespan on postural sway for each group independently. We found that within Group A, there was no relationship between COP displacement and fractional lifespan, while in Group B, RMS Ov. increased with the increase in fractional lifespan. These results suggest a non-linear relationship between aging and postural sway, with balance unaffected by age in orthopedically and neurologically normal dogs during most of their life, but with an onset of postural control decline in senior dogs. Studies in humans have also demonstrated a non-linear decline in balance and vestibular function with parameters staying steady up to 50 to 60 years of age (approximately 70% of the average lifespan) [46–48].

Fractional lifespan was the best variable to explain the variability in RMS Ov while Acc CC did not seem to be influenced by aging. Therefore, aging in dogs appears to have less effect on postural control in the CC direction, while being important in the overall movement. In this regard, in older humans, the sense of balance in the ML direction is thought to be more significant in preventing falling than in the CC direction. Additionally, in both humans and cats, it has been established that ML balance control depends only on the hip abductor and adductor muscle activity, while CC control can occur within multiple joints (hip, ankle, and knee) [49–51]. According to this, it has been proposed that the higher impact of aging on ML than on CC balance might be due to the fact that aging is associated with muscle atrophy and ML postural control relies on fewer muscle groups [11, 52]. In this work, COP measurements in the ML

direction were not the most useful measures to differentiate younger from older dogs, however, the fact that the overall displacement seems to be more affected than the CC suggests that aging might be affecting both CC and ML, as well as other angular directions.

## Role of pain, owners' perception of function, and proprioception in postural sway

Pain showed a major influence on postural sway in the CC direction in older dogs. We evaluated pain severity by two different methods; owners' perception of pain severity by means of the Pain Severity subscale of the CBPI and clinical evaluation during the orthopedic examination. Additionally, we evaluated the owner's perception of their dogs' function/mobility by means of the Pain Interference subscale of the CBPI and the LOAD questionnaire. Pain evaluated by a clinician was correlated with both COP measurements, having the greatest effect on Acc CC. According to these results, total joint pain seems to be critical in the craniocaudal displacement of the COP. One of the most common reasons for joint pain is osteoarthritis (OA), and the risk of clinical OA has been reported as increasing with age in dogs [53]. In our study we found a positive linear relationship between total joint pain and fractional lifespan, showing that older dogs are more likely to have greater joint pain, however this relationship was not statistically significant. Another study has demonstrated that COP displacement increased in dogs with hip or elbow osteoarthritis in comparison with healthy dogs [23]; however, that study only estimated each paw COP instead of the whole-body COP and they evaluated the pain at each joint individually while we considered the sum of the pain in all joints. Osteoarthritis has also been associated with higher COP displacement in humans, being even higher in painful OA [54–57]. In this regard, it has been demonstrated in humans that osteoarticular pain can decrease muscle strength as well as proprioceptive acuity [54, 58]. In dogs with OA, muscle atrophy is a consistent finding [53, 59]. Further studies including muscle condition scores, and muscle strength are needed to determine how these factors might be contributing to balance impairment in dogs.

It is interesting to note that the owners' perception of pain and function were not correlated with postural sway. The owner's perception of pain was evaluated by the Pain Severity subscale of the CBPI, and while questions are completely focused on pain severity, they do not specify the origin of pain. Hence, joint pain specifically might be more important in postural control than pain from other origin. Additionally, each owner may have a different way of evaluating pain in their dogs increasing the variability of the data, while total joint pain was evaluated always by the same veterinarian.

Unexpectedly, we did not observe differences in Acc CC or RMS Ov. between normal dogs and dogs with proprioceptive deficits. However, the interaction between total joint pain and proprioceptive deficits did play a role in Acc CC. The correlation coefficient between total joint pain and Acc CC was much higher (and significant) in dogs with CP deficits than normal dogs. In this regard, Lewis *et al.* (2019) found an increase in both CC and ML displacement of COP in dogs with chronic thoracolumbar spinal cord injury in comparison with neurologically normal dogs [24]. However, that study evaluated dynamic rather than static posturography and used non-ambulatory paraparetic dogs. In humans, decreased balance was also found in patients with proprioceptive deficits, and proprioceptive inputs are considered to play a critical role in postural control [60, 61]. In our study, the lack of clear differences between dogs with normal and abnormal proprioception could be related to the fact that only 9 (22.5%) of the Group B dogs had placing response deficits, and these were mild, with none of them having absent placing responses in any of the limbs. Additional studies are needed to evaluate COP displacement during quiet standing in dogs with more severe CP deficits.

## Several factors influence COP displacement

Various studies have demonstrated that multiple factors play a role in balance control and isolating the effect of each variable independently becomes challenging. It is known that aging is associated with an increased risk of developing joint pain [62] and a reduction in proprioceptive acuity [63, 64]. Moreover, chronic joint diseases can impair proprioception due to damage to mechanoreceptors [65, 66]. In this study we have determined that total joint pain is related with acceleration of the COP in the craniocaudal direction; and this is more marked in dogs with CP deficits. Therefore, part of the difference in postural stability between younger and older dogs might be mainly due to joint pain and proprioceptive deficits. However, fractional lifespan was the only relevant variable that explained variance in RMS Ov., suggesting that other comorbidities associated with aging not included in this study, such as muscle wasting, could have a more important role in the overall displacement of the COP.

## Measurements to use in future studies

This study intended to be a preliminary exploration of COP measurements to identify which measurements would be most useful for future studies. Because these measurements are highly correlated, using more than one might be redundant [20, 67]. In this study using a stepwise regression we selected two independent variables that produced a model able to discriminate between younger and older dogs with excellent accuracy. Of note, the correlation between these two variables was one of the lowest, indicating that they are assessing slightly different characteristics of postural sway. Additionally, they belong to different groups of variables: Acc reflects the length of the sway path, while RMS Ov is dependent on the extension of the COP displacement. It has been proposed that extension measures show the effectiveness of the postural control system (i.e., in an effective postural control, COP does not approach the limits of the base of support), while path measures represent the amount of regulatory activity needed to achieve that postural control [20, 67, 68]. Because of this, we consider that future studies should include at least one sway path variable and one extension variable. In our study Acc showed high reliability and was affected by total joint pain and its interaction with CP deficits. On the other hand, RMS was less able to detect significant changes produced by increases in fractional lifespan and was less affected by pain making it a good measure to detect age-associated changes not explained by joint pain or proprioceptive impairment. However, the selection of variables could depend on the specific goal of the experimenter or clinician, and therefore future studies might select different variables.

## Limitations

This study has several limitations; first, we did not evaluate the effect of other comorbidities associated with aging such as muscle atrophy/sarcopenia or vestibular dysfunction. Additionally, the evaluation of the role of neurological impairments that affect locomotion was limited in this study because all dogs were able to walk independently and had very mild proprioceptive deficits, if any. Further studies with a larger number of dogs are needed to determine the most important health issues associated with aging have the greatest impact on balance. However, it might not be possible to collect this kind of data in dogs who have severe difficulties standing still for several seconds. Another limitation is that while we used the fractional lifespan to avoid differences in aging between different sizes of dogs, this does not consider the different pace of aging that each individual dog may have. It would be interesting to evaluate the effect of frailty in postural sway in dogs at the same fractional lifespan. Finally, as this was an exploratory study, we did not correct for multiple comparisons, future studies should select a subset of COP measurements and use these types of corrections.

In conclusion, our study has demonstrated for the first time that static posturography using PSW is a reliable tool to evaluate postural sway in dogs during quiet standing. Additionally, postural sway appears to depend on several factors. It increases with age and is highly affected by one of the main comorbidities in older dogs, joint pain. Finally, considering its reliability, and accuracy to discriminate between younger and older dogs, and to avoid redundancy in measures, Acc CC and RMS Ov appear to be the best measures of COP displacement to capture the effect of joint pain and aging in dogs respectively.

## Supporting information

**S1 File. Clinical evaluation sheets used in the study.** This file contains the physical, neurologic, and orthopedic examination sheets that were used to record clinical examination. (PDF)

**S2 File. Raw data.** Data used in this manuscript can be found in this file. (XLSX)

## Acknowledgments

The authors are grateful with all the owner's that agreed to have their dogs participate in this study. We are grateful with Beth Case from the Department of Clinical Sciences, NCSU for her technical assistance.

## Author Contributions

**Conceptualization:** Alejandra Mondino, Natasha Jane Olby.

**Data curation:** Alejandra Mondino.

**Formal analysis:** Alejandra Mondino, Grant Wagner, Katharine Russell, Edgar Lobaton, Emily Griffith.

**Investigation:** Katharine Russell.

**Methodology:** Alejandra Mondino, Edgar Lobaton, Margaret Gruen.

**Project administration:** Natasha Jane Olby.

**Resources:** B. Duncan X. Lascelles.

**Supervision:** Edgar Lobaton, Natasha Jane Olby.

**Writing – original draft:** Alejandra Mondino.

**Writing – review & editing:** Grant Wagner, Katharine Russell, Edgar Lobaton, Emily Griffith, Margaret Gruen, B. Duncan X. Lascelles, Natasha Jane Olby.

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
