## [Decision Letter · Decision Letter 0]

24 Mar 2022

PONE-D-22-04659Static posturography as a novel measure of the effects of aging on postural control in dogsPLOS ONE

Dear Dr. Olby,

Thank you for submitting your manuscript to PLOS ONE. After careful consideration, we feel that it has merit but does not fully meet PLOS ONE’s publication criteria as it currently stands. Therefore, we invite you to submit a revised version of the manuscript that addresses the points raised during the review process.

The method presented in your study is interesting, novel, and potentially very useful to the veterinary community. Thank you for submitting it to PLOS ONE. The reviewers liked your manuscript and the overall approach of the study. One reviewer requested additional attention to the statistical analysis---please be sure to address those questions. Please submit your revised manuscript by May 08 2022 11:59PM. If you will need more time than this to complete your revisions, please reply to this message or contact the journal office at plosone@plos.org. Please include the following items when submitting your revised manuscript:A rebuttal letter that responds to each point raised by the academic editor and reviewer(s). You should upload this letter as a separate file labeled 'Response to Reviewers'.A marked-up copy of your manuscript that highlights changes made to the original version. You should upload this as a separate file labeled 'Revised Manuscript with Track Changes'.An unmarked version of your revised paper without tracked changes. You should upload this as a separate file labeled 'Manuscript'.

We look forward to receiving your revised manuscript.

Kind regards,

Richard Evans

Academic Editor

PLOS ONE

Journal Requirements:

2. In your Methods section, please provide additional details regarding participant consent from the owners of the animals. In the ethics statement in the Methods and online submission information, please ensure that you have specified (1) whether consent was informed and (2) what type you obtained (for instance, written or verbal). If the need for consent was waived by the ethics committee, please include this information.

Reviewers' comments:

Reviewer's Responses to Questions

**Comments to the Author**

1. Is the manuscript technically sound, and do the data support the conclusions?

Reviewer #1: Yes

Reviewer #2: Yes

2. Has the statistical analysis been performed appropriately and rigorously? 

Reviewer #1: I Don't Know

Reviewer #2: No

3. Have the authors made all data underlying the findings in their manuscript fully available?

Reviewer #1: Yes

Reviewer #2: No

4. Is the manuscript presented in an intelligible fashion and written in standard English?

Reviewer #1: Yes

Reviewer #2: Yes

5. Review Comments to the Author

Reviewer #1: Static posturography as a novel measure of the effects of aging on postural control in dogs.

This is a reasonably well written paper that generated a large volume of data. Interpretation of that data is challenging because the findings are not always uniform. The authors discussion of limitations and final conclusions are fair, but some clarifications are needed. My biggest obstacles with this paper were 1) interpreting how they selected data, 2) the shear volume of acryonyms and data presented will challenge many readers, 3) ICC is calculated on averaged data, not raw data and 4) the number of statistical evaluations performed.

67 “…cheaper…” this is brand dependent and certain brands are at least twice the cost of a force platform set up. Please remove.

91/442 Why 0.75 as a cutoff, why not the 0.7 cutoff as supported by this cited literature. Also, since time is a continuous variable, why not evaluate the data to see if there is change in COP associated with adjusted age. Regardless, at least provide rationale for making this cutoff. It may be as easy as this was exploratory research and other options beyond a 0.75 cutoff should be investigated.

98/409 Was this the extent of the morphometric measurements? Why dog height and not length and width? Not investigating if other morphometric measurements influence COP is a potential limitation of the study.

131 Is this the definition of a valid trial?

136 Do you mean 70 consecutive frames, from 7 trials? I’m unsure of what you did here given your previous description of inclusion criteria; were some dogs studied even though they didn’t achieve the “inclusion criteria”. Were these frames and trials randomly selected? In the discussion, you mention you averaged all of this data. Please clarify how you handled this data, it is unclear to me and I would not be able to reproduce this experiment. From the review of your data, it seems that you began with 5360 data points, from this you selected (random?)(and averaged?) 70 data points from 7 of 10 (random?) valid trials leaving your with 7 data points for each dog, then this was averaged. Is this what was statistically evaluated? Post hoc handling of your data needs to be clear and logical.

137 On line 129, you stated, “for inclusion, at least 10 valid trials were required”. Clarify how you could have data if “fewer than 7 individual trials were available”.

222 It seems this should be calculated on the raw data, not averaged data that is a subset of the raw data. Please defend this methodology or perform on raw data.

Tables Often, the r^2 are significant, but quite small. Please address the importance of these findings.

Reviewer #2: First, I will make some background comments and then suggest specific changes to the manuscript. Do not feel obliged to respond to the background comments, but you can if you want to. I provided them to put my specific recommendations in context.

Background comments

I approached this paper as someone who might use this method to assess the efficacy of a therapy using a clinical trial and COP measures.

What would I need to know from your paper to help me design and analyze my study?

1. I would like to see summary statistics (e.g., means, standard deviations, etc.) of the COP outcomes, for each group. That way, I could put my results in the context of your results, albeit informally. If my numbers were very different from yours, I might have a problem with my study or population.

2. I would like to know the COP outcomes with the lowest relative variation, assessed with their coefficients of variations. That way, I can directly compare variables. This is a simple, standard approach.

3. I would like to know which variables and combination of variables best distinguish between group A and group B dogs. That way, I can pick a couple of primary variables before my study begins. Finding such variables is commonly done with an ROC curve analysis. I understand that lines 508+ attempts to suggest variables, but correlations don't tell me what I want to know. An area under the ROC curve does.

This manuscript mainly reported results from regressions. Regressions have four assumptions: independence, linearity, conditional normality, and homoscedasticity.

It's not clear that the investigators checked any of these in the analysis. Note that the normality of POS outcomes themselves is not one of the linear regression assumptions.

The manuscript primarily reports three statistics: the adjusted r-square, the standardized beta, and p-values. The adjusted r-squares and the standardized beta aren't very helpful--even the authors don't use them to describe the results in the papers. They are simply put in tables and left there.

The authors mostly use statistical significance to assess their results. The problem is that there are over 221 p-values assessed for statistical significance in this report. That means we expect to see more than 11 statistically significant results that are false discoveries. But we don't know which are false discoveries. (221x0.05 is about 11)

So when the authors rely on statistical significance to make claims such as line 407 and line 428, we don't know if they are among the 11 false-positive statements.

Another problem with the p-values is that they depend on sample size so that a weak effect can become statistically significant with a larger sample size. For example, height has a minimal effect (r-square) on COP variables. Frankly, I'd ignore height in this analysis of COP data. But because your sample size is large enough to make height statistically significant, it appears to have more importance than it deserves.

Also, a particular COP variable might have the same effect size in both groups, but because group B has a smaller sample size than group A, there might be statistical significance in A but not B. So, the disparity in sample sizes can affect group comparison based on p-values

Finally, I challenge you to remove the lines and colors in figure 4 and see if your eye can detect changes in postural sway with fractional age. The problem is that the human eye focuses on the lines but not really on the confidence intervals. I could draw a legitimate line within the intervals on the group B side that angles downwards, showing that postural sway decreases with fractional age. In fact, you should have statistically compared the slopes of the A and B sides of those plots. If those slopes are different, then age might affect sway.

Specific suggested changes--major points

1. Please consider reporting my additional analyses noted in points 1 to 3 above. Before suggesting those analyses, I tried them on your data. It took only an hour, and I was able to discount height as a factor--its slope is tiny. I also found that two COP variables differentiate group A and group B well, with AUC=0.8. (I used a stepwise logistic regression).

2. I'm not sure how you can claim that older dogs have worse sway because (1) groups A and B were never directly compared with any statistical test, and (2) group B is confounded with lame dogs. In particular, I'm not sure the statements in lines 435+ are correct. It looks to me like a couple of outlying dogs are driving your statements.

Please compare the groups directly on COP variables and the regression slopes in figure 4, and explain how you untangle the lameness (pain, etc.) issues.

4. Discuss the adjusted r-square and standardized betas in the discussion section or remove them.

6. If you wish to keep the regressions, then check the regression assumptions for each regression.

Minor points

1. I believe you used multivariate regression but sometimes called it multiple regression, which is different. Please fix the terminology.

2. Put Dr. Lascelles name on one line. The "B." is at the end of the previous line.

6. PLOS authors have the option to publish the peer review history of their article (what does this mean?). If published, this will include your full peer review and any attached files.

Reviewer #1: No

Reviewer #2: No

---

## [Author Response · Author response to Decision Letter 0]

24 Apr 2022

Dear Dr. Evans,

 Thank you for the careful reviews that were performed on our manuscript entitled “Static posturography as a novel measure of the effects of aging on postural control in dogs” by Mondino et al., (Manuscript ID: PONE-D-22-0465 for publication in PLOS One. We have submitted the revised version in which we have attended to the Reviewers' comments, the requests have been incorporated into the revised manuscript, and we consider that the manuscript is greatly improved. We hope the reviewers feel we have addressed their comments and concerns adequately. We also added a statement about the approval of all protocols by our IACUC committee and that all owners reviewed and signed an informed consent form. 

Reviewer #1

Comment: This is a reasonably well written paper that generated a large volume of data. Interpretation of that data is challenging because the findings are not always uniform. The authors discussion of limitations and final conclusions are fair, but some clarifications are needed. My biggest obstacles with this paper were 1) interpreting how they selected data, 2) the shear volume of acryonyms and data presented will challenge many readers, 3) ICC is calculated on averaged data, not raw data and 4) the number of statistical evaluations performed.

Response: We thank the reviewer for taking the time to review our manuscript and providing us with valuable comments and suggestions which have helped us to improve the clarity of the manuscript and its technical aspects.

We have clarified how the data was selected (see below), we have removed one of the acronyms (TJP is now presented as total joint pain). We have also reduced the amount of data and statistical evaluation performed as we selected by means of stepwise regression only the variables that explained best the variation in postural stability between younger and senior dogs. 

Regarding the ICC, I think there is a misunderstanding about how this was calculated. We didn’t calculate the ICC on averaged data, we used the data of each of the seven periods individually, but we selected the ICC analysis type “mean of k”. According to Koo and Li (2016), the type of ICC needs to be selected according to the measurement protocol that will be conducted in the actual application. In this case, this type of ICC analysis was selected because we planned to use the mean value of 7 different measurements for further analysis. As we now explained in the manuscript, this approach has also been used in several human studies.

Comment: 67 “…cheaper…” this is brand dependent and certain brands are at least twice the cost of a force platform set up. Please remove.

Response: The word cheaper has been removed.

Comment: 91/442 Why 0.75 as a cutoff, why not the 0.7 cutoff as supported by this cited literature. Also, since time is a continuous variable, why not evaluate the data to see if there is change in COP associated with adjusted age. Regardless, at least provide rationale for making this cutoff. It may be as easy as this was exploratory research and other options beyond a 0.75 cutoff should be investigated.

Response: We based the cutoff on the AAHA guidelines, and they consider a dog senior at the last 25% of the estimated lifespan through the end of life. Therefore over 75%, not 70%. 

We did compare the effect of fractional lifespan in each group of dogs individually, and we saw that the relationship is not linear through the entire life, as in dogs younger than 75% of their lifespan there is no correlation between fractional lifespan and postural stability. As it happens with several health parameters, there is an onset of decline in the senior population. (This was also demonstrated in humans). 

Comment: 98/409 Was this the extent of the morphometric measurements? Why dog height and not length and width? Not investigating if other morphometric measurements influence COP is a potential limitation of the study.

Response: We agree with the reviewer’s comment, and we have now included the data of length. All the three morphometric variables that we evaluated (height, length and weight) were highly correlated, so we don’t think that adding width will make a huge difference. 

Comments: 131 Is this the definition of a valid trial?

Response: We have clarified how we selected valid (acceptable) frames. We visually review the videos and selected the frames where dogs were standing square, completely still and looking forward. Those were considered “acceptable frames”. 

136 Do you mean 70 consecutive frames, from 7 trials? I’m unsure of what you did here given your previous description of inclusion criteria; were some dogs studied even though they didn’t achieve the “inclusion criteria”. Were these frames and trials randomly selected? In the discussion, you mention you averaged all of this data. Please clarify how you handled this data, it is unclear to me and I would not be able to reproduce this experiment. From the review of your data, it seems that you began with 5360 data points, from this you selected (random?)(and averaged?) 70 data points from 7 of 10 (random?) valid trials leaving your with 7 data points for each dog, then this was averaged. Is this what was statistically evaluated? Post hoc handling of your data needs to be clear and logical.

Response: We thank the reviewer for this comment. We have clarified this in the methods section of the manuscript and hope that it now reads in a manner that allows reproduction. 

At least 10 trials were recorded for each individual dog. Following acquisition, videos were visually reviewed and only the frames where the dogs were standing square, completely still and looking forward were considered acceptable for analysis. We selected 7 periods of 70 acceptable frames to use in further analysis. 

If at least 7 trials had periods of 70 acceptable frames, the script selected one period of each trial in the order they were recorded, if not, it sampled the trials available more than once and selected additional non-overlapping periods.

COP variables were calculated for each period of 70 frames, and then values were averaged to obtain one value per dog for each COP variable.

Comment: 137 On line 129, you stated, “for inclusion, at least 10 valid trials were required”. Clarify how you could have data if “fewer than 7 individual trials were available”.

Response: As explained above, this was clarified.

Comment: 222 It seems this should be calculated on the raw data, not averaged data that is a subset of the raw data. Please defend this methodology or perform on raw data.

Several similar studies in humans use averaged data, because the variability of each individual trial is high. In fact, there are some studies evaluating how many trials need to be averaged to obtain a high reliability measure. We have clarified this in the manuscript.

We selected the ICC type “mean of k measures” because our intended protocol averages the results of several trials, and not using just one trial. This decision was based in several studies in humans that have demonstrated that an average of a minimum of two trials is needed to obtain a reliable measure of postural stability

A detailed explanation on how to select the ICC analysis can be found in Koo and Li (2016) paper, cited in our manuscript.

Comment: Tables Often, the r^2 are significant, but quite small. Please address the importance of these findings.

Response: We think that each individual parameter evaluated have some contribution to postural stability, even if it doesn’t explain a higher % of the variation. We have now modified the statistical analysis. We have now focused the analysis on trying to find the most relevant variables than could explain variation in postural stability, rather than looking at each individual variable that was correlated with it. We have been helped by a statistician (now included as a co-author), and we think that this new approach has significantly improved the manuscript. 

Reviewer #2: 

Comments: First, I will make some background comments and then suggest specific changes to the manuscript. Do not feel obliged to respond to the background comments, but you can if you want to. I provided them to put my specific recommendations in context.

Background comments

I approached this paper as someone who might use this method to assess the efficacy of a therapy using a clinical trial and COP measures.

What would I need to know from your paper to help me design and analyze my study?

1. I would like to see summary statistics (e.g., means, standard deviations, etc.) of the COP outcomes, for each group. That way, I could put my results in the context of your results, albeit informally. If my numbers were very different from yours, I might have a problem with my study or population.

Response: We thank the reviewer for taking the time to make a thorough revision of our manuscript and providing us with useful comments and suggestions, which we have addressed in the revised version.

In the previous version we showed summary statistics using bar graphs, however we now show them in Table 4, so it is easier for the reader to look at the exact median and range values. 

Comment: 2. I would like to know the COP outcomes with the lowest relative variation, assessed with their coefficients of variations. That way, I can directly compare variables. This is a simple, standard approach.

Response: While we didn’t show coefficient of variations, we have evaluated the reliability of each COP measurement using an Intraclass correlation (ICC) analysis. Also, the range of each variable is now shown in Table 4. 

Comment: 3. I would like to know which variables and combination of variables best distinguish between group A and group B dogs. That way, I can pick a couple of primary variables before my study begins. Finding such variables is commonly done with an ROC curve analysis. I understand that lines 508+ attempts to suggest variables, but correlations don't tell me what I want to know. An area under the ROC curve does.

Response: We are grateful with the reviewer for this comment. We have changed our statistical analysis, and we have selected the variables that explained variation between young and old dogs better. We have evaluated the accuracy of the model including those variables to differentiate between the two groups of dogs by means of a ROC curve. 

Comment: This manuscript mainly reported results from regressions. Regressions have four assumptions: independence, linearity, conditional normality, and homoscedasticity.

It's not clear that the investigators checked any of these in the analysis. Note that the normality of POS outcomes themselves is not one of the linear regression assumptions.

Response: We checked these assumptions and as several variables were not normally distributed we log-transformed them to achieve normality.

Comment: The manuscript primarily reports three statistics: the adjusted r-square, the standardized beta, and p-values. The adjusted r-squares and the standardized beta aren't very helpful--even the authors don't use them to describe the results in the papers. They are simply put in tables and left there.

The authors mostly use statistical significance to assess their results. The problem is that there are over 221 p-values assessed for statistical significance in this report. That means we expect to see more than 11 statistically significant results that are false discoveries. But we don't know which are false discoveries. (221x0.05 is about 11)

So when the authors rely on statistical significance to make claims such as line 407 and line 428, we don't know if they are among the 11 false-positive statements.

Another problem with the p-values is that they depend on sample size so that a weak effect can become statistically significant with a larger sample size. For example, height has a minimal effect (r-square) on COP variables. Frankly, I'd ignore height in this analysis of COP data. But because your sample size is large enough to make height statistically significant, it appears to have more importance than it deserves.

Also, a particular COP variable might have the same effect size in both groups, but because group B has a smaller sample size than group A, there might be statistical significance in A but not B. So, the disparity in sample sizes can affect group comparison based on p-values

Response: We are thankful for these comments, and we have changed the statistical analysis, which we hope has improved this manuscript. We now focused on only two variables that explained the differences between younger and older dogs, and we used stepwise regression to determine which variables were most relevant in postural stability. We have used R2 to show how much of the variation in one variable was explained by the independent variable. We used standardized beta, so the reader can compare the effect size of each variable even when the variables are measured in different units.

Comment: Finally, I challenge you to remove the lines and colors in figure 4 and see if your eye can detect changes in postural sway with fractional age. The problem is that the human eye focuses on the lines but not really on the confidence intervals. I could draw a legitimate line within the intervals on the group B side that angles downwards, showing that postural sway decreases with fractional age. In fact, you should have statistically compared the slopes of the A and B sides of those plots. If those slopes are different, then age might affect sway.

This is a very interesting point, and we have now compared the slopes of the curves showing that age affects sway in the senior population but not in the younger dogs.

Specific suggested changes--major points

Comment: Please consider reporting my additional analyses noted in points 1 to 3 above. Before suggesting those analyses, I tried them on your data. It took only an hour, and I was able to discount height as a factor--its slope is tiny. I also found that two COP variables differentiate group A and group B well, with AUC=0.8. (I used a stepwise logistic regression).

Response: We are grateful with the reviewer for taking the time to perform this analysis. We have now changed the statistical analysis as suggested.

Comment: I'm not sure how you can claim that older dogs have worse sway because (1) groups A and B were never directly compared with any statistical test, and (2) group B is confounded with lame dogs. In particular, I'm not sure the statements in lines 435+ are correct. It looks to me like a couple of outlying dogs are driving your statements.

Please compare the groups directly on COP variables and the regression slopes in figure 4, and explain how you untangle the lameness (pain, etc.) issues.

Response: In the previous version we have compared each individual variable between young and older dogs and most of them were significant. This was shown in the older Figure 3. However, now, we changed the statistical analysis trying to find the best variables able to differentiate between groups. We also evaluated the effect of FLS in each group of dogs individually and compared the regression slopes, showing that for RMS Ov, there is a significant difference in the slope between groups.

We are aware that pain could be responsible for the differences between the groups, because of this we evaluated the relationship between pain, proprioception and fractional lifespan, and postural sway. Using a stepwise regression analysis, we found that while pain and proprioceptive deficits were relevant for acceleration of the COP, fractional lifespan was the only relevant variable that explained the variation on RMS Ov.

Comment: Discuss the adjusted r-square and standardized betas in the discussion section or remove them.

Response: Thank you for this suggestion, adjusted R2 have been removed. Standardize Beta was kept because it shows the effect size of each variable, and it allows to compare between variables with different units.

Comment: If you wish to keep the regressions, then check the regression assumptions for each regression.

Response: Assumptions were checked, and variables that were not normally distributed were log transformed.

Minor points

1. I believe you used multivariate regression but sometimes called it multiple regression, which is different. Please fix the terminology.

Response: Thank you for noticing this terminology mistake. This has been modified.

2. Put Dr. Lascelles name on one line. The "B." is at the end of the previous line.

Response: Done

---

## [Editor Report · Decision Letter 1]

29 Apr 2022

Static posturography as a novel measure of the effects of aging on postural control in dogs

PONE-D-22-04659R1

Dear Dr. Olby,

We’re pleased to inform you that your manuscript has been judged scientifically suitable for publication and will be formally accepted for publication once it meets all outstanding technical requirements.

Kind regards,

Richard Evans

Academic Editor

PLOS ONE

Additional Editor Comments (optional):

Thank you for your timely revision of the manuscript. We know it was a lot of work, and we appreciate your professionalism.
---

## [Editor Report · Acceptance letter]

16 Jun 2022

PONE-D-22-04659R1 

Static posturography as a novel measure of the effects of aging on postural control in dogs 

Dear Dr. Olby:

I'm pleased to inform you that your manuscript has been deemed suitable for publication in PLOS ONE. Congratulations! Your manuscript is now with our production department. 

Kind regards, 

on behalf of

Dr. Richard Evans 

Academic Editor

PLOS ONE